# Public Feeding Interactions as Enrichment for Three Zoo-Housed Elephants

**DOI:** 10.3390/ani11061689

**Published:** 2021-06-06

**Authors:** Eduardo J. Fernandez, Bruce Upchurch, Nancy C. Hawkes

**Affiliations:** 1School of Animal and Veterinary Sciences, The University of Adelaide, Adelaide, SA 5005, Australia; 2Woodland Park Zoo, Seattle, WA 98103, USA; bupchurch@comcast.net (B.U.); nancy.hawkes@zoo.org (N.C.H.)

**Keywords:** animal welfare, animal–visitor interactions, elephants, enrichment, guest feeding, human–animal interactions, public feeding, stereotypy, visitor feeding, zoo

## Abstract

**Simple Summary:**

Human–animal interactions are an important focus of modern animal welfare research. A subset of this interest includes animal–visitor interactions that occur in zoos. One understudied aspect of animal–visitor interactions involves public feedings, where visitors can directly feed the zoo animals. We examined the effects of public feedings compared with nonpublic feed days on the general activity of three zoo-housed elephants. In addition, we examined the general activity of the elephants in the months prior to public feedings, as well as their general activity on public feed days before, during, and after a public feeding. Public feedings were effective at increasing social activity and decreasing stereotypies for two of the elephants when compared with nonpublic feed days. Additionally, all three elephants showed increased foraging and decreased inactivity following a public feeding. Our results suggest that public feedings can be an effective form of environmental enrichment for zoo-housed elephants.

**Abstract:**

The past few decades have seen increased interest in studies examining the welfare of elephants and animal–visitor interactions. One understudied area for both pursuits is the impact of public feeding interactions. Our study examined the effects of public feedings on the general activity of three zoo-housed elephants. Prior to public feedings, we developed and assessed a 21-behavior ethogram split into six classes of behavior. Comparisons between the elephants demonstrated that only one of the elephants engaged in stereotypies with regularity (>30%), and that the stereotypies occurred in place of most foraging. During public feedings, we compared the general activity of each elephant independently and across both public feeding and nonpublic feeding days, as well as the general activity before, during, and after a public feeding. Public feedings increased social activity and decreased stereotypies when compared with nonpublic feeding days for two of the elephants. In addition, all three elephants showed increased foraging and decreased inactivity in the period after a public feeding session. These results demonstrate that public feedings can be a useful tool for enriching the welfare of zoo-housed elephants and are among the first sets of data to demonstrate positive welfare outcomes associated with public feedings.

## 1. Introduction

Over the past several decades, zoos have considerably increased their efforts to assess and improve the welfare of their animals through research and management practices [1,2,3]. Two species of note for these efforts have been African (*Loxodonta africana*) and Asian (*Elephas maximus*) elephants, with initial evaluations suggesting lower zoo survivorship for both species [4,5,6,7], as well as non-self-sustaining zoo populations for Asian elephants [8,9,10,11]. The above findings resulted in the Association of Zoos and Aquariums (AZA) and similar organizations (e.g., The British and Irish Association of Zoo and Aquariums (BIAZA) and the European Association of Zoos and Aquaria (EAZA)) standardizing behavioral welfare tools and prioritizing multi-institutional elephant research [12,13,14,15,16,17,18,19]. As a result, numerous epidemiological, multi-institutional studies on zoo-housed elephants have examined various welfare-related factors, including the use of environmental enrichment, social grouping and conspecific contact, housing, feeding and foraging activity, elephant–keeper interactions, husbandry training and management, and stereotypic activity [20,21,22,23,24,25,26,27]. These efforts have allowed accredited zoos to create exhibits and practices that are presumed to improve the housing and management of both African and Asian elephants.

Similar welfare-oriented research efforts have examined the effects of animal–visitor interactions in zoos, documenting both positive and negative effects of those interactions [28,29,30,31,32]. One type of animal–visitor interaction includes public feedings, where zoo visitors can directly deliver food to a zoo-housed animal [33]. Concerns over the use of public feeding interactions include potential animal or visitor injury and non-natural perceptions of such interactions [34,35]. Nonetheless, researchers have noted that visitors actively enjoy such interactions [36,37,38,39], while others have documented little to no adverse effects on the animals because of public feeding opportunities [40,41,42,43]. In addition, factors related to visitor–elephant interactions, such as observing an interactive elephant program or entering an immersive elephant exhibit area, have been associated with greater visitor enjoyment and/or conservation support [44,45]. However, only two published studies to date have experimentally examined the effects of public feeding interactions compared with a nonpublic feeding control condition on any zoo-housed species, finding increased keeper interactions and no detrimental welfare effects on crowned lemurs (*Eulemur coronatus*) and few behavioral differences in domestic chickens (*Gallus gallus*), respectively [46,47].

The following study examined the effects of visitor public feeding interactions on three zoo-housed elephants. Woodland Park Zoo decided to introduce zoo visitor feedings primarily as an opportunity for up-close interactions that invoked awe for elephants while supporting their mission to inspire people to “learn, care, and act” for wildlife. Twenty-one behaviors were split into six classes of behavior and examined for both between- and within-elephant differences. Prior to the start of public feedings, we evaluated the overall activity patterns of each elephant and compared them across all three elephants. During the public feedings, we documented the behavioral effects of public feeding days compared with nonpublic feeding days on each elephant independently, as well as the behavioral effects observed before, during, and after a public feeding. We hypothesized that (1) in the observations prior to public feedings, differences in stereotypic activity between the three elephants would be directly correlated with differences in individual foraging activities, (2) public feedings would increase social and foraging activities and decrease stereotypies when compared with nonpublic feeding days, and (3) the majority of any positive behavioral effects observed on a public feeding day would occur in the periods during and after a public feeding.

## 2. Materials and Methods

### 2.1. Subjects and Enclosure

Three elephants were the subjects of the study: Bamboo, a female Asian elephant, ~3800 kg; Chai, a female Asian elephant, ~3750 kg; and Watoto, a female African elephant, ~3700 kg. Bamboo was estimated to be 43 years old at the start of the study, Chai was 30 years old at the start of the study, and Watoto was estimated to be 40 years old at the start of the study. All three elephants resided at Woodland Park Zoo (Seattle, WA, USA) since 1 January 1968 for Bamboo, 30 March 1980 for Chai, and 1 July 1971 for Watoto.

The elephants resided in an exhibit in the Asian Tropical Forest biome of the zoo, which contained multiple areas, all on view to the public: an indoor elephant house, ~200 m^2^; a northern hill area, ~1250 m^2^; a middle pool area, ~500 m^2^, which contained a pool with ~ 105 kL of water; and a southern yard area, ~1700 m^2^ (see Figure 1). All outdoor areas consisted of or were surrounded by natural trees, deadfall, grass, and sand. The perimeter of the exhibit was intentionally designed and maintained to provide live browsable plant material for the elephants year-round. Feeding enrichment was routinely provided in all areas and in the form of scatter feeds and through devices, such as hanging feeders, boomer balls, and fixed plastic drum feeders. The middle pool area contained several rows of seating for the public to watch keeper talks and/or interactions. The indoor elephant house contained public viewing windows and individual spaces for each elephant. The indoor elephant barn also functioned as a location to perform routine husbandry care in front of guests, such as foot baths/trims, as well as a night house where the elephants could be held overnight. The elephants were frequently given free access to several areas of the exhibit for the entire day. Depending on weather and temperature conditions (<5 °C), the elephants were held in the indoor elephant house overnight and moved to the outdoor areas of their exhibit from 10:00 to 17:00 h (fall/winter, October–March) or 08:30 to 18:30 h (spring/summer, April–September).

Diets for the elephants varied based on both the individual and time of year, with 4–4.5 kg of produce, 35–45 kg of grass hay, ad lib browse, including cut bamboo, 2–3 kg of organic elephant supplements, Rovimix^®^ (vitamin E, DSM, Inc., Heerlen, the Netherlands), and 1.5 tsp of plain, granular salt consumed per elephant per day. Produce per elephant per day consisted of ~1500 g apple slices, ~1000 g yam slices, ~1000 g carrot pieces, ~250 g quartered melons, and ~500 g whole bananas. Approximately half of the total daily diet was delivered throughout the day and in the form of scatter feeds, hidden on exhibit, placed in various enrichment devices (see above), or used as rewards during husbandry training procedures and/or keeper interactions. The remaining half of total daily diet was split into three deliveries: mornings (08:30–10:00 h), afternoons (13:00–15:00 h), and evenings (17:00–18:30 h).

### 2.2. Materials

Materials included the food items used during the public feedings, which were taken from the afternoon portion of an elephant’s daily diet. Other materials included Palm^®^ handhelds (Palm, Inc., Sunnyvale, CA, USA) used to record behavioral data, an Event-PC program that was run on the Palm^®^ handhelds and designed specifically for this experiment by Dr. James C. Ha at the University of Washington, and a notebook used to record potential errors and additional observations/field notes that occurred during a session.

### 2.3. Design and Procedure

Prior to its implementation, the study was approved through Woodland Park Zoo’s Research Committee, as well as the University of Washington’s Institutional Animal Care and Use Committee (IACUC #2858-06). An ethogram consisting of 21 behaviors split into 6 classes of behaviors was developed for use with all observations (see Table 1).

The behaviors observed were mutually exclusive, and the inclusion of the “Other” observation category made the ethogram exhaustive. An instantaneous time (pinpoint) sampling procedure [48] was used to record behaviors for each elephant independently during all observation sessions. The Event-PC program would randomly assign an elephant to observe for a 5 min session, and observations were recorded for that elephant every 15 s (20 samples per session). Observers were typically scheduled to observe in 1 h increments, with 7–9 observation sessions conducted per hour.

All observations were conducted between 08:30 and 18:30 h, 7 days a week, between 20 August 2009 and 27 September 2011. There were two distinct periods of observation during this time: (1) prior to the public feedings (20 August 2009–21 February 2010), where we assessed the average daily activity of each elephant (1381 total observation sessions for ~115 h of observation), and (2) during public feedings (3 May 2011–27 September 2011), where we assessed the effects of both public feeding and nonpublic feeding days on each elephant (1549 total observation sessions for ~129 h of observation; see details below). These two observation periods were distinct in that no comparisons were made between the two periods of observation (all latter public feeding versus nonpublic feeding comparisons were based on the second period of observation; see Methods below).

All public feedings occurred between 13:30 and 14:30 h in the middle pool area of the exhibit, with the normal afternoon feeding for all 3 elephants occurring after a public feeding. The schedule for public feedings was determined at least 1 week ahead of time, with 1–2 elephants scheduled during any 1 public feeding day, and each elephant scheduled for a public feeding at least twice a week. During a public feeding, individuals or small groups of visitors were selected to approach a keeper and elephant from the public viewing area. Keepers would then hand a visitor a 1 m pole with browse from the normal diet fixed to the end of the pole and allow the elephant to use its trunk to collect the food item (see Figure 2 and Figure 3).

All visitors were given the opportunity to pay to participate in the public feeding, each public feeding event occurred for 60 min, and any food items not used were added back to the afternoon diet for that elephant. For all public feeding comparisons, public feedings (Public Feed (PF) condition) were directly compared with days in which each elephant did not have a public feeding (No Public Feed (No PF) condition) within the same period (3 May 2011–27 September 2011; PF = 532 total sessions, No PF = 1017 total sessions). Therefore, a Public Feed (PF) day for 1 or 2 elephants was a No Public Feed (No PF) day for both or 1 of the other elephants, respectively.

A total of 28 observers collected data for the entire study. Observers were typically registered for independent research credit through the Psychology Department at the University of Washington (PSY 499) and received observation training by live training sessions at the beginning of each semester and weekly lab meetings throughout the study. Observations were examined weekly by the first author for consistency across all observers, and any deviations were accounted for during these weekly checks, as well as through weekly lab meetings. In the latter public feeding portion of the study, observers could identify public feeding events for each elephant when a public feeding was occurring but were otherwise unaware of when public feedings occurred and were unfamiliar with the expected behavioral outcomes of the study.

### 2.4. Statistical Analyses

SigmaStat^®^, version 11.0 (Systat Software Inc., San Jose, CA, USA), was used to run all statistical analyses. Because Shapiro–Wilk tests for normality and/or equal variance tests failed, nonparametric tests were used to conduct all statistical analyses. Kruskal–Wallis analysis of variance (ANOVA) on ranks tests were used to examine prior to public feeding differences in the classes of behaviors and for select behaviors (see Results) between the three elephants. When significant differences (*p* < 0.05) for the ANOVAs were found, post hoc pairwise comparisons (using Dunn’s method) were implemented. During the public feedings, differences between the public feeding (Public Feed (PF) condition) and nonpublic feeding (No Public Feed (No PF) condition) days for each elephant were tested using Mann–Whitney U tests, with the number of sessions per condition directly compared. Finally, for the public feeding days for each elephant, Kruskal–Wallis analysis of variance (ANOVA) on ranks tests were used to examine differences in the periods before (8:30–12:59 h), during (13:00–14:59 h), and after (15:00–18:30 h) the public feeding. When significant differences (*p* < 0.05) for the ANOVAs were found, post hoc pairwise comparisons (using Dunn’s method) were implemented.

## 3. Results

### 3.1. Prior to Public Feedings: Differences between Elephants

Prior to the start of public feedings (20 August 2009–21 February 2010), we examined differences in activity between all three elephants. Figure 4 shows differences in the six classes of behaviors across Bamboo, Chai, and Watoto.

Three of the classes of behaviors tested for differences between the three elephants showed a statistically significant effect: Forage (*x*^2^_2_ = 70.814, *p* < 0.001), Inactive (*x*^2^_2_ = 56.581, *p* < 0.001), and Stereotypy (*x*^2^_2_ = 318.289, *p* < 0.001). For all three classes, post hoc tests showed a significantly lower occurrence in the Forage and Inactive classes of behavior and a significantly higher occurrence in the Stereotypy class of behaviors when comparing Chai with Bamboo and Watoto (*p* < 0.05 for all).

To further examine the relationship between foraging and stereotypies, we compared differences in the occurrence of three of the behaviors in the Forage class (Feeding, Foraging Other, and Enrichment Feeding) and the two behaviors in the Stereotypy class (Rocking and Pacing) across all three elephants. Figure 5 shows the differences in the five behaviors between Bamboo, Chai, and Watoto.

All five behaviors tested for differences across the three elephants showed a statistically significant effect: Feeding (*x*^2^_2_ = 14.817, *p* < 0.001), Foraging Other (*x*^2^_2_ = 69.870, *p* < 0.001), Enrichment Feeding (*x*^2^_2_ = 48.353, *p* < 0.001), Pacing (*x*^2^_2_ = 6.248, *p* = 0.044), and Rocking (*x*^2^_2_ = 571.526, *p* < 0.001). For Feeding behaviors, post hoc tests showed a significantly higher occurrence for Bamboo when compared with Chai and Watoto (*p* < 0.05 for both). For Foraging Other behaviors, post hoc tests showed a significantly lower occurrence for Chai when compared with Bamboo and Watoto (*p* < 0.05 for both). For Enrichment Feeding behaviors, post hoc tests showed a significantly higher occurrence for Watoto when compared with Bamboo and Chai (*p* < 0.05 for both). For Rocking behaviors, post hoc tests showed a significantly higher occurrence for Chai when compared with Bamboo and Watoto (*p* < 0.05 for both). For Pacing behaviors, there were no post hoc differences observed between the three elephants.

### 3.2. Public Feed Days Compared with No Public Feed Days

To examine the effects of public feedings, we compared differences in activity between the day of a public feeding (Public Feed (PF) condition) and all other days (No Public Feed (PF) condition) during the same period (3 May 2011–27 September 2011) and for each elephant independently. Figure 6 shows the differences in all six classes of behaviors between the PF and No PF conditions for Bamboo (top graph), Chai (middle graph), and Watoto (bottom graph).

For Bamboo, there was a significant increase in the Social class of behaviors (*U*_502_ = 47,202.500, *p* < 0.001) and the Inactive class of behaviors (*U*_502_ = 37,780.000, *p* < 0.001), and a significant decrease in the Forage class of behaviors (*U*_502_ = 50,680.500, *p* < 0.001) when comparing the PF condition with the No PF condition. For Chai, there was a significant increase in the Social class of behaviors (*U*_522_ = 45,167.500, *p* = 0.028) and a significant decrease in the Stereotypy class of behaviors (*U*_522_ = 39,086.000, *p* = 0.005) when comparing the PF condition with the No PF condition. For Watoto, there was a significant increase in the Forage class of behaviors (*U*_519_ = 54,709.500, *p* = 0.009) and a significant decrease in the Stereotypy class of behaviors (*U*_519_ = 48,090.500, *p* = 0.011) when comparing the PF condition with the No PF condition.

### 3.3. Before, During, and After a Public Feeding

To further examine the effects of public feeding interactions, we compared differences in activity on days in which a public feeding occurred between the periods of time before (8:30–12:59 h), during (13:00–14:59 h), and after (15:00–18:30 h) a public feeding for each elephant independently. This was examined by taking the occurrence of each class of behavior during that time and subtracting from it the baseline activity (No Public Feed (No PF) condition) for that same behavioral class and elephant. For instance, if an elephant displayed on average 50% of the Forage class of behaviors during the No PF (baseline) days, and then exhibited 25%, 50%, and 75% of the Forage class of behaviors before, during, and after a public feeding, this would be presented as a −25%, 0%, and +25% change from baseline, respectively. Figure 7 shows the differences in the change from baseline activity in all six classes of behaviors in the periods before, during, and after a public feeding for Bamboo (top graph), Chai (middle graph), and Watoto (bottom graph).

For Bamboo, three of the classes of behaviors showed a statistically significant effect: Forage (*x*^2^_2_ = 35.025, *p* < 0.001), Social (*x*^2^_2_ = 19.925, *p* < 0.001), and Inactive (*x*^2^_2_ = 44.043, *p* < 0.001). For the Forage class of behaviors, post hoc tests showed a significantly higher occurrence in the After period when compared with the Before and During periods (*p* < 0.05 for both). For the Social class of behaviors, post hoc tests showed a significantly higher occurrence in the During period when compared with the Before and After periods (*p* < 0.05 for both). For the Inactive class of behaviors, post hoc tests showed a significantly lower occurrence in the After period when compared with the Before and During periods (*p* < 0.05 for both).

For Chai, four of the classes of behaviors showed a statistically significant effect: Forage (*x*^2^_2_ = 31.535, *p* < 0.001), Social (*x*^2^_2_ = 6.488, *p* = 0.039), Inactive (*x*^2^_2_ = 13.482, *p* = 0.001), and Stereotypy (*x*^2^_2_ = 23.568, *p* < 0.001). For the Forage class of behaviors, post hoc tests showed a significantly higher occurrence in the After period when compared with the Before and During periods (*p* < 0.05 for both). For the Inactive class of behaviors, post hoc tests showed a significantly lower occurrence in the After period when compared with the Before and During periods (*p* < 0.05 for both). For the Stereotypy class of behaviors, post hoc tests showed a significantly lower occurrence in the After period when compared with the Before and During periods (*p* < 0.05 for both). For the Social class of behaviors, there were no post hoc differences observed between the three periods.

For Watoto, four of the classes of behaviors showed a statistically significant effect: Active (*x*^2^_2_ = 14.910, *p* < 0.001), Forage (*x*^2^_2_ = 28.946, *p* < 0.001), Social (*x*^2^_2_ = 13.332, *p* = 0.001), and Inactive (*x*^2^_2_ = 37.652, *p* < 0.001). For the Active class of behaviors, post hoc tests showed a significantly higher occurrence in the Before period when compared with the During and After periods (*p* < 0.05 for both). For the Forage class of behaviors, post hoc tests showed a significantly higher occurrence in the After period when compared with the Before and During periods (*p* < 0.05 for both). For the Inactive class of behaviors, post hoc tests showed a significantly lower occurrence in the After period when compared with the Before and During periods (*p* < 0.05 for both). For the Social class of behaviors, there were no post hoc differences observed between the three periods.

## 4. Discussion

### 4.1. Prior to Public Feedings: Foraging and Stereotypies

Prior to the start of public feedings, we examined the overall activity of all three elephants and found that one of the three elephants, Chai, exhibited higher occurrences of stereotypies than the other two elephants (see Figure 4: Chai, 33.7% (*SE* = 1.7); Watoto, 8.2% (*SE* = 1.1); Bamboo, 6.3% (*SE* = 0.9)). This higher occurrence of stereotypic activity was correlated with lower foraging and inactivity for Chai when compared with the other two elephants. To further examine this pattern, we compared three of the foraging responses and both stereotypies across all three elephants as well (see Figure 5). Watoto engaged in more Enrichment Feeding than the other two elephants, Bamboo in more Feeding than the other two elephants, and Chai in less Foraging Other and more Rocking than the other two elephants. Simply stated, Bamboo spent much of her foraging time eating food delivered by the keepers (Feeding), Watoto spent much of her foraging time eating food delivered in enrichment devices (Enrichment Feeding), and both elephants would otherwise spend much of their time foraging from trees or bushes found in and around their exhibit (Foraging Other). Chai appeared to spend less time in any of these foraging activities, and alternatively spent most of her time engaged in Rocking. Additionally, Rocking accounted for >95% of all of Chai’s stereotypic activity, as opposed to ~4% and ~6% of Watoto’s and Bamboo’s stereotypic activities, respectively.

Previous research has demonstrated appetitive functions to stereotypies in zoo animals [49,50,51,52,53,54,55,56]. Likewise, Rees [57] found that increased feeding opportunities resulted in decreased stereotypic activity in Asian elephants. While other factors have been positively correlated with stereotypies in elephants, such as time spent alone, time spent indoors, lower temperatures, and winter seasons [23,58,59], these results suggest that Chai’s Rocking behavior replaced or otherwise had an appetitive foraging function. These results also provided a clear rationale for examining public feeding opportunities with the zoo’s elephants: public feeding interactions could provide extra foraging opportunities for Chai, who otherwise spent less of her time foraging on her own.

### 4.2. Public Feeding General Effects

Public feedings (PF) were effective in reducing stereotypic activity in two of the three elephants (Chai and Watoto) when compared with the nonpublic feeding (No PF) days. In addition, public feedings increased the social activity in two of the three elephants (Bamboo and Chai) and increased foraging in one of the elephants (Watoto) when compared with nonpublic feeding days. What was less clear is why Bamboo showed significantly more inactivity and less foraging during public feeding interactions compared with nonpublic feedings, although this is partially explained in relation to the effects observed in the period after a public feeding (see below). Regardless, overall, the public feedings appeared to have positive behavioral welfare effects on the elephants.

Most of the increased social activity observed (>90%) involved keeper interactions (Interacting with Keeper), which was expected and consistent with a prior study examining the effects of public feeding interactions on zoo-housed crowned lemurs [46]. Similarly, Keulen-Kromhout [60] found that public feedings were correlated with increased attempted visitor interactions and lower occurrences of stereotypies in zoo-housed bears. Increased elephant–keeper interactions have also been positively correlated with increased welfare for zoo elephants and increased keeper satisfaction [21]. Additionally, visitors often seek animal interactions, and such interactions can be a source of enrichment for some zoo animals [28,29,32]. Therefore, aside from the welfare benefits of increasing foraging and decreasing stereotypic activity, public feedings can foster both greater positive animal–visitor and animal–keeper interactions, which has the potential to further enhance the welfare of zoo animals.

### 4.3. Before and After a Public Feeding

In the periods of time that occurred before and after a public feeding, all three elephants significantly increased their foraging and decreased their inactivity after a public feeding interaction. In addition, Bamboo showed significant increases in social activity during a public feeding, Chai showed significant decreases in stereotypic activity after a public feeding, and Watoto showed significant increases in Active behaviors before a public feeding. These results suggest that the public feedings had their greatest effect on all three elephants in the period after a public feeding. In other words, public feedings themselves appeared to elicit general foraging activity for all three elephants. For Bamboo, who of the three elephants spent the most time eating food directly delivered by the keepers (see above and Figure 5), this effect was large enough to increase overall inactivity (i.e., waiting) and decrease foraging before and during a feeding, as well as when comparing public feedings with nonpublic feedings (see above and Figure 6). The effects of any environmental events, including public feeding interactions, should be examined in terms of both their occurrence (i.e., treatment) and nonoccurrence (i.e., baseline), as well as what happens before and after those events. In the case of the elephants in our study, the greatest welfare effects appeared to occur after a public feeding, which could easily have been dismissed as a nonbeneficial effect on Bamboo if only considering the occurrence and nonoccurrence of a public feeding.

While caution is warranted in extrapolating these findings beyond this limited context, the results suggest that some aspect of public feeding interactions may function as enrichment for some zoo elephants. Future research could focus on examining the predictability and type of interactions that occur during public feedings. Previous research has demonstrated that making feeding times less predictable or providing extra feeding opportunities has beneficial welfare effects [61,62,63,64,65]. Likewise, public feedings could employ enrichment device training and introductions, thus increasing enrichment usage while educating the public about the importance of enrichment [66]. Public feedings could prove to be a useful tool for enhancing the welfare of zoo-housed elephants while simultaneously offering visitors engaging interactions with keepers and elephants.

## 5. Conclusions

Historically, public feedings have played an important role in the animal–visitor interactions possible for zoo visitors. However, little has been done to empirically examine the effects of public feedings. Our study effectively demonstrated decreased stereotypies and increased social activity as a result of public feedings and in contrast to nonpublic feeding days for two of the three elephants. In addition, all three elephants decreased inactivity and increased foraging in the period after a public feeding. These results are among the first to document positive welfare benefits for zoo-housed animals due to public feeding interactions. Zoos should further examine the potential benefits of public feedings for a variety of their exhibited animals, with respect to both the welfare benefits for those animals and the increase of positive human–animal interactions that can help achieve their conservation education missions.

## Figures and Tables

**Figure 1 animals-11-01689-f001:**
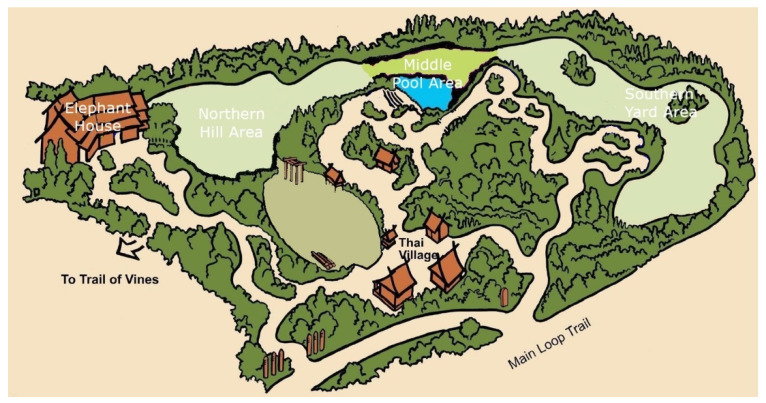
Elephant exhibit, with all four major areas listed (elephant house, northern hill area, middle pool area, and southern yard area).

**Figure 2 animals-11-01689-f002:**
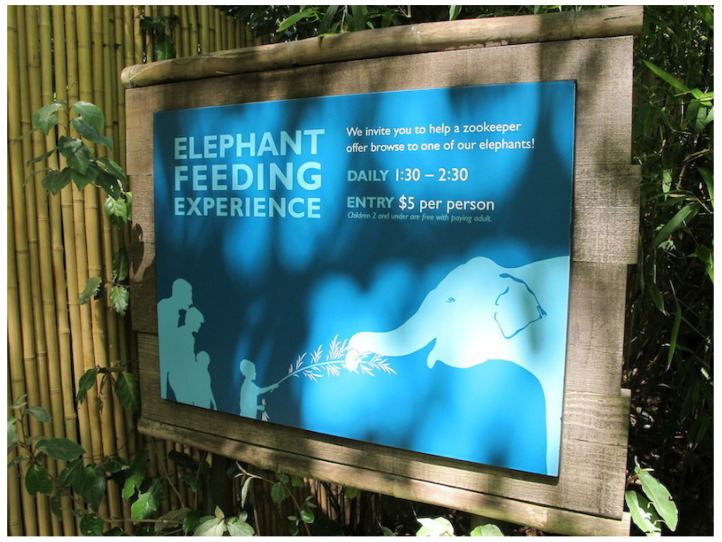
Signage for the public feedings. Photo credit: Kristen Pisto and Woodland Park Zoo.

**Figure 3 animals-11-01689-f003:**
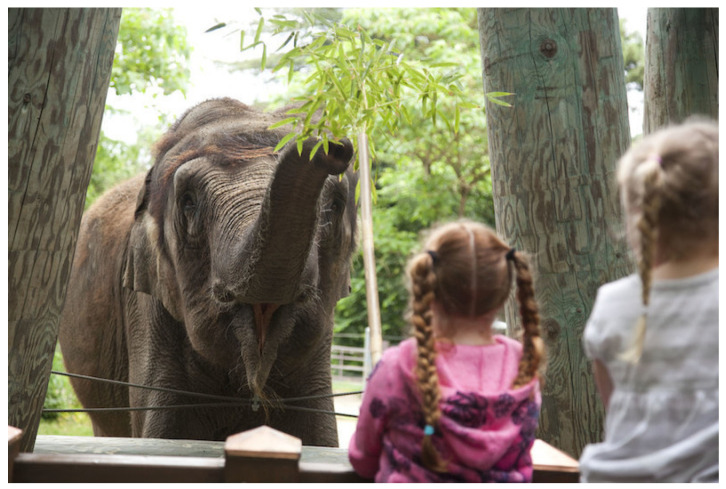
Chai and two visitors during a public feeding interaction. Photo credit: Lauren LaPlante and Woodland Park Zoo.

**Figure 4 animals-11-01689-f004:**
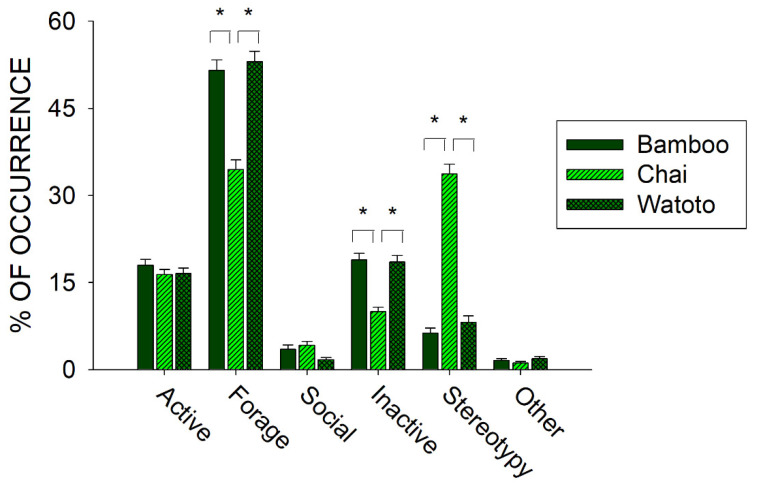
Comparison of the average activity (with standard error of the mean) of each of the six classes of behavior across all three elephants in the period of observations prior to the public feedings. Lines plus * designate statistically significant differences within a class of behavior and between elephants (*p* < 0.05).

**Figure 5 animals-11-01689-f005:**
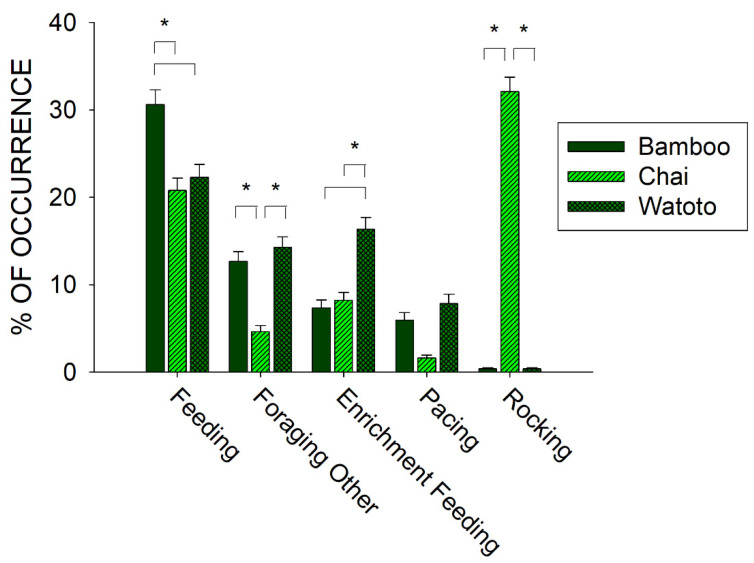
Comparison of the average activity (with standard error of the mean) of five of the behaviors in the Forage and Stereotypy classes of behaviors and across all three elephants in the period of observations prior to the public feedings. Lines plus * designate statistically significant differences within a behavioral category and between elephants (*p* < 0.05).

**Figure 6 animals-11-01689-f006:**
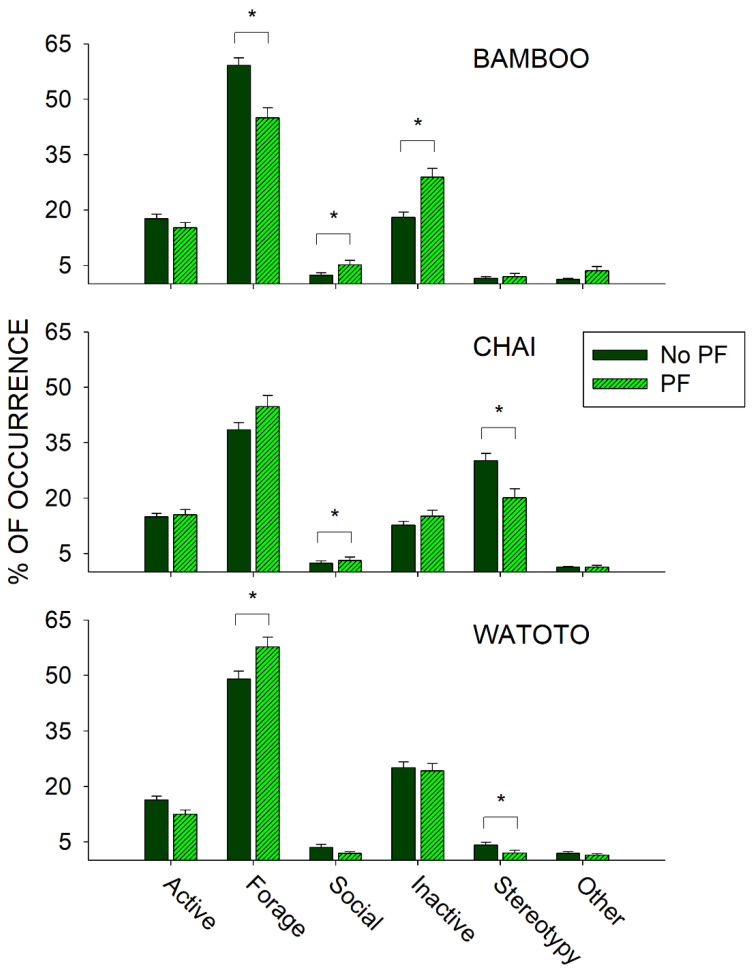
Comparison of the average activity (with standard error of the mean) of each of the six classes of behavior and all three elephants for the No Public Feed (No PF) and Public Feed (PF) conditions. Lines plus * designate statistically significant differences for each elephant within a behavioral category and between the No PF and PF conditions (*p* < 0.05).

**Figure 7 animals-11-01689-f007:**
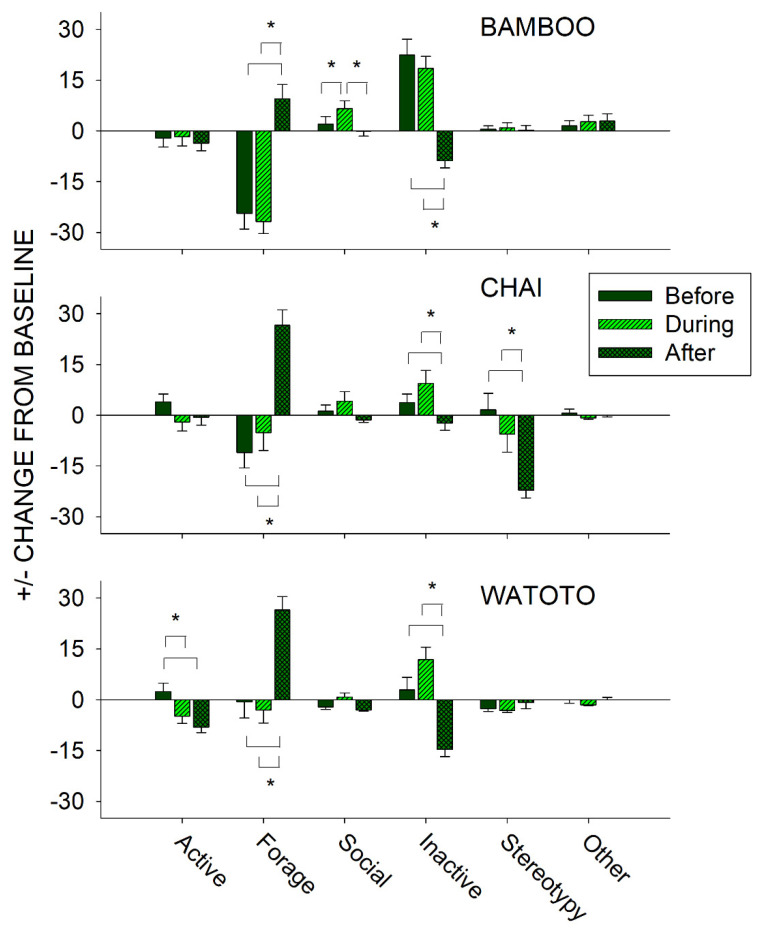
Comparison of the change from baseline activity (with standard error of the mean) of each of the six classes of behavior and all three elephants for the periods of time that occurred before, during, and after a public feeding. Lines plus * designate statistically significant differences for each elephant within a behavioral category and between the before, during, and after periods (*p* < 0.05).

**Table 1 animals-11-01689-t001:** Behaviors, classes of behavior, and definitions for each response in the ethogram.

Behavioral Class and Behaviors (Abbreviations)	Definitions
Active	
Enrichment Contact (EC)	Manipulating any nonfood enrichment item.
Locomotion (Lo)	Directed nonrepetitive movement.
Digging/Dusting (DD)	Manipulating substrate or throwing substrate on body.
Play/Bathe (PB)	Splashing, kicking, bouncing, or other animated behavior (must be alone).
Forage	
Feeding (FE)	Eating any food provided by the keepers.
Foraging Other (FO)	Reaching over the fence, browsing on exhibit shrubs, or grazing on exhibit grasses.
Enrichment Feeding (EF)	Eating any item inside an enrichment device.
Drinking (Dr)	Mouth contact with pool or other water source.
Social	
Affiliation (Af)	Conspecific, prosocial behaviors (e.g., play, trunk tangle, or caressing with elephant).
Interacting with Keeper (IK)	Any keeper-related interaction.
Inactive	
Lying Down (LD)	At least two limbs no longer upright (e.g., on side).
Standing (St)	Completely upright with no movement.
Gate Hugging (GH)	Standing, but in front of a gate and without movement.
Stereotypy	
Pacing (Pa)	Moving in a repetitive pattern, with completion from point A to B and back to point A (must include at least one full A–B–A movement) or circling.
Rocking (Ro)	Moving back and forth without locomotion. Must include at least one full back-and-forth motion.
Other	
Vocalization (Vo)	Making nondirected vocal sound.
Social Aggression (AG)	Aggressive behavior that includes physical contact.
Threat/Display (TD)	Aggressive behavior, but without physical contact.
Urinate/Defecate (UD)	Urination or defecation.
Out of Sight (OS)	Not visible to the observer.
Other (Ot)	Engaged in a behavior not listed above.

## Data Availability

The data presented in this study are available on request from the corresponding author.

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
