# Peer review of "Public Feeding Interactions as Enrichment for Three Zoo-Housed Elephants"

_animals, 2021, doi:10.3390/ani11061689_

Round 1

Reviewer 1 Report

Dear Authors,

attached please find a pdf with comments.

Regards

Author Response

We have adjusted the title as well as the first sentence of the introduction. Per the comment about statistical analyses, detail is provided in the methods section, and it is common practice within animal behavior research to run inferential statistics with few subjects per condition. The question is more about external validity (generalizability) rather than the ability to implement statistical analyses, and we have included the following sentence in the discussion which notes some of the concern over external validity: "While caution is warranted in extrapolating these findings beyond this limited context, the results suggest that some aspect of public feeding interactions may function as enrichment for some zoo elephants."

Reviewer 2 Report

This study evaluates the impact of public feeding on welfare in 3 zoo-housed elephants of 2 species. The study was conducted over a decade ago (lines 142-143)! The delay in publishing is a bit surprising, to say the least. The results should, in principle, still be valid, given the nature of the question. I have several concerns with the analyses and overall interpretation of the results.

The salient feature of public feedings is very likely to be the increased opportunity for  elephants to interact with their keepers. That this interaction is occurring during a public feeding is potentially irrelevant. Keeper presence could be the real salient feature, not the public feeding aspect per se. It would be ideal to compare the results to simply having more keeper interaction on some days. I suspect this would lead to similar positive results (though would not bring in $$, as I imagine the public feeding interactions were for a fee). I would suggest including more details in this regard about the public feeding program.

Lines 102-103: I assume “foot baths and trims” should be “foot trims and baths”

Line 172 – it sounds as if no formal  inter observer reliability was conducted (that is, “Observations were examined weekly by the first author for consistency across all observers”). This seems questionable, at best. Some indication of reliability – percent agreement at least – should be included.

Analyses: it does not appear that P values were adjusted for multiple comparisons. Obviously, an increase in the percent time spent in 1 behavior necessitates a decrease in other behaviors. Using P<.01 in most cases seems prudent.

Can you clarify what your df was for analyses, and include this in the results? Is your n 3 elephants? Or is it based on number of observations? Is an observation a 5-min observation, an hour, or a day?

Figures 1 and 2 are based on 2009 data, and figure 3 is based on the 2011 data. It seems that there was a noticeable decrease in pacing between these 2 observation periods, irrespective of whether it was a public feed day. Might other changes in husbandry or environment explain this decrease?

Line 335: “Public feedings (PF) were effective in reducing stereotypic activity in two of the three elephants.” I’d be careful of over-stating the results. Only 1 of the 3 elephants engaged in frequent stereotypy. The frequency was so low in the other elephants that saying there was a significant reduction in 2 of the 3 is a bit overstated. It clearly made a huge difference for 1 of the 3 elephants, at least.  The same could be said for the change in social behavior: overall, it was a very rare behavior (5% or less of the time). To say that public feeding increased social behavior is also a bit overstated, given how infrequently it occurred irrespective of public feeding.

While the results are interesting, I think the role of public feedings could be greatly overstated. Without a comparison of behavior when keeper interaction is increased (without visitor interaction with the elephants), it is hard to really conclude that public feedings have a positive impact. Greater keeper interaction has a positive impact; the presence of visitors could be secondary. Such a comparison could readily be made, with additional data collection. How valid that would be 10 years later is of course a concern.

Author Response

We have included our responses as an attached file.

Reviewer 3 Report

This was a well designed and conducted study. The analysis is thorough and the presentation is accessible and easy to understand. The significant benefit to the elephants of this easy and effective intervention is an important finding for all captive elephants. The additional benefit to the zoo visiting public and their experience with exotic animals is also potentially important although that was not assessed in the current study.

Author Response

Thank you.

Reviewer 4 Report

Overview:  Very well-written paper with high value

Abstract: The abstract and simple summary are easy to understand for both lay and scientific audiences with clear and concise description of population studied as well as procedures and outcomes. 

Introduction:  Given the limited research in this specific area, it is understandable that the introduction is slightly shorter than normal and feels appropriate to the subject.  The project goals are clearly described and good context is provided for the study. 

Materials and Methods: 

Good description of environment and what was already accessible in terms of environment, routines, interactions, feeding, and enrichment options. 

Individual researchers and observation techniques are clearly described with statistical procedures also clearly described.

Results:    Clear description of what resulted from the statistical analysis and well-presented in graphs.  It might be helpful to change lines 277 – 305 into a table for easier reading. 

Discussion:  There is great discussion here about the effects of public feeding on elephant behavior and how that might correlate to positive welfare.  Regarding pro-social behavior, it might be good to include some points regarding how increased pro-social behavior can also indicate higher levels of stress (neither good nor bad) which can be a function of increased arousal due to enrichment in PF days or potentially as a coping mechanism for potential higher levels of distress. 

Author Response

Thank you. We opted not to include an extra table, as we decided to include an extra figure of the enclosure and wanted to limit including much more information. Also, in addressing some of Reviewer 2's points, we included the following sentence, which notes some of the difficulties in generalizing our results beyond this study: "While caution is warranted in extrapolating these findings beyond this limited context, the results suggest that some aspect of public feeding interactions may function as enrichment for some zoo elephants." 

Hopefully, the above statement will cover the plethora of interesting topics related to public feeding interactions, including relevance to social behavior, that is further needed to better understand any potential benefits.

Round 2

Reviewer 2 Report

The authors have adequately addressed my concerns. I still wonder about the 10 year gap between data collection and publication, but that does not negate the validity of the results.